# Unraveling the Intricacies of Autophagy and Mitophagy: Implications in Cancer Biology

**DOI:** 10.3390/cells12232742

**Published:** 2023-11-30

**Authors:** Sunmi Lee, Ji-Yoon Son, Jinkyung Lee, Heesun Cheong

**Affiliations:** 1Branch of Molecular Cancer Biology, Division of Cancer Biology, Research Institute, National Cancer Center, Goyang-si 10408, Republic of Korea; 76992@ncc.re.kr (S.L.); jiyoon1095@ncc.re.kr (J.-Y.S.); 2Department of Cancer Biomedical Science, Graduate School of Cancer Science & Policy, National Cancer Center, Goyang-si 10408, Republic of Korea; dlwlsrud1992@ncc.re.kr

**Keywords:** autophagy, mitophagy, cancer, tumor microenvironment (TME), cancer-associated fibroblasts (CAFs), tumor-associated immune cells

## Abstract

Autophagy is an essential lysosome-mediated degradation pathway that maintains cellular homeostasis and viability in response to various intra- and extracellular stresses. Mitophagy is a type of autophagy that is involved in the intricate removal of dysfunctional mitochondria during conditions of metabolic stress. In this review, we describe the multifaceted roles of autophagy and mitophagy in normal physiology and the field of cancer biology. Autophagy and mitophagy exhibit dual context-dependent roles in cancer development, acting as tumor suppressors and promoters. We also discuss the important role of autophagy and mitophagy within the cancer microenvironment and how autophagy and mitophagy influence tumor host–cell interactions to overcome metabolic deficiencies and sustain the activity of cancer-associated fibroblasts (CAFs) in a stromal environment. Finally, we explore the dynamic interplay between autophagy and the immune response in tumors, indicating their potential as immunomodulatory targets in cancer therapy. As the field of autophagy and mitophagy continues to evolve, this comprehensive review provides insights into their important roles in cancer and cancer microenvironment.

## 1. Comprehensive Autophagic Processes

### 1.1. Autophagy in Normal Physiology

Macroautophagy (hereafter referred to as autophagy) is an intracellular catabolic pathway that operates through lysosomal degradation under conditions of nutrient deprivation or stress. For example, autophagy is frequently initiated in response to multiple stressors, such as oxygen deprivation, energy or amino acid deficits, or exposure to radiation and cytotoxic drugs. Unnecessary organelles, such as misfolded proteins, dysfunctional organelles, and other cytoplasmic components, are enveloped and sequestered into vesicle membranes and subsequently catabolized by lysosomes. The resulting breakdown products are recycled to serve as molecular building blocks or energy sources [1].

Although initially viewed as a “bulk degradation” process triggered by starvation, recent findings indicate that autophagy may also function as a highly selective, quality control system that manages the levels of specific organelles and proteins. During nutrient deprivation, cytoplasmic components are enveloped and catabolized in a non-selective manner within autophagosomes prior to recycling. In contrast, selective autophagy targets particular substrates into autophagosomes using specific autophagy receptors. The common feature shared by selective autophagy receptors is the presence of an LC3-interacting region (LIR). LIRs enable these receptors to bind to members of the LC3/GABARAP protein family, thereby connecting substrates with autophagosomes [2].

Selective autophagy can remove a range of substances, including protein aggregates, unwanted organelles, and pathogens. Autophagy also plays an important role in the aging process. Studies have demonstrated that a decreased expression of autophagic proteins in aged tissues results in a reduction in autophagy. This phenomenon is observed during normal human brain aging and conditions, such as osteoarthritis, in which the autophagy machinery is downregulated. Thus, autophagy plays a vital role in maintaining cellular homeostasis [3,4].

Autophagic processes may be divided into several steps, which include induction, initiation of autophagic vesicle formation, nucleation of a phagophore structure, cargo loading by cargo adaptors, elongation and maturation of the phagophore into an autophagosome, autophagosome fusion with the lysosome, and the subsequent degradation and recycling of nutrients [5].

### 1.2. Autophagy Machinery at Distinct Steps of the Process

There are approximately 40 autophagy-related proteins that have been identified, each playing a role in various steps of the process [6]. Autophagy-related genes (Atgs) were initially discovered in yeast and are highly conserved across lower eukaryotes and mammals.

To initiate autophagy, Unc-51-like kinase 1 (ULK1), which is the mammalian counterpart of the initially discovered autophagy-related gene 1 (Atg1), has an important role. ULK1 forms a complex with three other autophagy-related proteins: the 200 kDa focal adhesion kinase family-interacting protein (FIP200), ATG13, and ATG101. The interactions between ULK1 and ATG13 or FIP200 stabilize and enhance ULK1 kinase activity.

Moreover, ULK1 kinase activity can be modulated, either promoted or inhibited, by AMPK or the mTOR complex 1 (mTORC1) [7,8]. The ULK complex is typically inactive; however, it becomes active when mTORC1 is inhibited or when AMPK is activated as an upstream regulator. Consequently, the ULK complex integrates nutrient and energy stress signals to bridge the response of two key regulators, mTOR (anabolic) and AMPK (catabolic) [9]. Both AMPK and mTORC1 regulate the autophagy pathway through site-specific phosphorylation of ULK1 to attenuate its activity in an opposite manner [10]. Remarkably, recent studies suggest that in addition to ULK, other kinases, such as tank binding kinase 1 (TBK1), facilitate the formation of the ATG13–FIP200 protein complex by phosphorylating Syntaxin17 [11] (Figure 1).

Upon activation, the ULK1 kinase activates the Beclin1 (BECN1)–VPS34 complex, which consists of BECN1, VPS34 (a class III PI3K), and various other proteins, including VPS15, ATG14L, and autophagy and Beclin1 regulator 1 (AMBRA-1), with the composition varying based on the subcellular location of the complex [12]. The VPS34 lipid kinase complex prepares the membrane for curvature by producing phosphatidylinositol 3-phosphate (PI3P) on membranes, typically in the endoplasmic reticulum (ER).

A sequence of protein–lipid conjugation events links proteins, specifically those from the LC3 family, to the lipid membranes of the autophagosomes [13]. This modification serves to identify vesicles as autophagosomes and facilitate cargo reception. This process relies on the interaction between WIPI2B and PI3P, which is necessary for recruiting and organizing the two separate systems required for autophagosome formation.

In addition to its role in autophagosome elongation, LC3 also functions as a docking site for autophagic cargo receptors, marking autophagic cargo and facilitating its entry into the autophagosome (autophagic vesicle; AV). Cargo receptors, such as SQSTM1 (p62) and the neighbor of BRCA1 (NBR1), attach to proteins and organelles that have been tagged for autophagic degradation via ubiquitin markers [14]. Cargo receptors confer the ability of precision of autophagy by enabling specific cargoes to selectively attach to particular receptor molecules [15] (Figure 1).

Next, ubiquitin-like (UBL) proteins enlarge the phagophore, a precursor form of the autophagosome. As part of the first ubiquitin-like protein system, Atg12 is conjugated to Atg5 through interactions with Atg7 and Atg10 (E1- and E2-like enzymes, respectively) [16]. Following Atg5-12 conjugation, which is essentially a constitutive process, it associates with Atg16L. Interaction with ATG16L facilitates the conjugation of the microtubule-associated protein 1 light chain (LC3) to phosphatidylethanolamine (PE), which also involves Atg7 and Atg3, both of which exhibit E2-like enzyme activity during the conjugation process. This modified form of LC3, known as LC3-II, is located on autophagosomal membranes, whereas LC3-I primarily resides in the cytosol. These successive conjugations and spatial arrangements result in the phagophore maturing into an autophagosome surrounding the cargo (Figure 1).

The additional membrane is transported to the developing autophagosome, known as an autophagic vesicle (AV), to seal the vesicle. Lipid membranes sourced from mitochondria, the plasma membrane, Golgi, or the endoplasmic reticulum are used to form autophagosomes through ATG9 [17,18]. Next, mature autophagosomes fuse with lysosomes, a process regulated by Rab GTPases, membrane-tethering assemblies (including the HOPS complex and VPS genes), and soluble N-ethylmaleimide-sensitive factor attachment protein receptors (SNAREs) [19]. Finally, the cargo molecules within the autolysosomes are broken down by lysosomal hydrolases with the assistance of a lysosomal nutrient transporter to support cell growth [20,21].

Understanding the molecular regulatory processes at distinct stages of autophagy is crucial for gaining insights into effective strategies for regulating autophagy in its role in the progression of various diseases, like cancer.

### 1.3. Dual Roles of Autophagy in Cancer

With respect to cancer, autophagy initially acts to restrict the early phases of tumor formation; however, in well-established tumors, autophagy manages internal and external stresses, including hypoxia, nutrient depletion, and response to treatment, ultimately promoting tumor progression. The autophagy process maintains a stable amount of protein and organelles by removing damaged proteins and small organelles digested by lysosomes while maintaining cell metabolism and survival under conditions of starvation and stress [22]. During the early stages of tumorigenesis, autophagy acts as a tumor suppressor by degrading damaged organelles or harmful substances and suppressing the spread of damage, including oxidative stress, DNA modification, and genome stability, thereby inhibiting tumorigenesis [23,24]. Because autophagy is associated with longevity, as DNA damage accelerates during cancer and aging, autophagy protects cellular and genome integrity to prevent cancer and extend lifespan [25] (Figure 2).

One representative example, Beclin-1/ATG6, is a crucial factor in autophagy initiation and functions as a tumor suppressor [26]. A higher incidence of spontaneous carcinomas and lymphomas in lung, liver, and breast tissues was observed in *beclin1^+/−^* mice compared with *beclin^+/+^* mice [27]. Moreover, the deletion of a single allele or epigenetic silencing of Beclin-1 is associated with 50–70% of human breast, ovarian, and prostate cancers [28]. These findings confer the significance of autophagy as a tumor suppressor during cancer progression.

The deletion of ATG5 results in a tumor-suppressive phenotype, akin to the presence of heterozygote mutation of BECN1 in cancer models, whereas only benign hepatomas are observed in a mouse liver model [29]. In patients with colorectal and gastric cancer, decreased autophagy has been observed resulting from mutations in ATG2, ATG5, ATG9, and ATG12. Furthermore, ATG12 inhibits the expression of the anti-apoptotic protein BCL-2, thereby inhibiting tumor cell survival [30]. In triple-negative breast cancer (TNBC), ATG2B and ATG5 act as suppressors of cancer stemness through the induction of autophagy [31]. Taken together, these findings suggest that autophagy has a tumor-suppressive role in various cancers.

Interestingly, hepatic adenomas occur in *Atg5^flox/flox^*, *CAG-Cre* and *Atg7^flox/flox^*, and Alb-Cre liver-specific Atg-deleted mice, but tumor size is reduced by deleting liver-specific p62. These results suggested that autophagy inhibition in early periods causes spontaneous tumor development in the liver, and the accumulation of p62 contributes to tumor progression [29]. P62 deficiency reduces tumor development caused by defective autophagy, indicating that abnormal p62 accumulation resulting from a deficiency in autophagy contributes to the early stages of cancer development. p62 expression promotes oxidative stress and tumor growth [32] and contributes to the growth and pathogenicity of clear cell renal cell carcinoma [33]. These results imply that p62 plays a critical positive role in early tumorigenesis in various cancers, regardless of autophagy activity.

During the advanced or late stages of tumorigenesis, however, autophagy plays a role as a tumor promoter. Although autophagy contributes to the maintenance of genome stability and prevents cellular damage during the initial stages of cancer progression, it impacts cancer cell survival through a strategy reminiscent of normal cells. This involves the preservation of functional mitochondria and the facilitation of intracellular recycling, thereby supplying the substrates necessary for metabolism within the demanding microenvironment encountered by cancer cells (Figure 2).

The cancer-promoting role of autophagy has been established using various mouse models. Several studies using genetically engineered mouse (GEM) models of cancer have revealed direct evidence supporting the cancer-promoting function of autophagy. The dependence of cancer on autophagy is supported by the observation of elevated autophagic activity accompanying the activation of the Ras oncogene. The presence of H-Ras or K-Ras oncogenic mutants results in an increase in basal autophagy, a process necessary for the survival of tumor cells under conditions of nutrient starvation and during tumorigenesis [34]. This role for autophagy in cancer can be established through the deletion of Atg7 in a spontaneous KRas*^G12D^*-driven non-small cell lung cancer (NSCLC) GEM model. In lung cancer mice with Atg7 or Atg5 deletions, tumor size decreased compared with wild type Atg7 or Atg5 mice [35,36].

An epithelial skin tumor was induced in mice by the epidermal keratinocyte-targeted deletion of Atg7 as follows. In the normal epidermis, 7, 12-dimethylbenz(a)anthracene (DMBA) was administered to induce HRas mutations, and 12-O tetradecanoylphorbol-13-acetate (TPA) was administered to generate tumors. In addition, K5-Son of sevenless (SOS) EGFR^wa2/wa2^ mice, in which wounding induces tumors, were used. In mouse tumors generated by chemical treatment, there was no significant change resulting from the deletion of Atg7, but the deletion of Atg7 in tumors caused by HRas mutation markedly suppressed tumor progression based on the size of the tumor and the survival rate of the mice [37].

Several studies focused on the downstream effector, BRAF, because Kras mutation-driven tumors are highly sensitive to autophagy inhibition. Mutations in BRAF resulted in the constitutive activation of the kinase and RAS independence, as a valine to glutamate replacement in amino acid residue occurs in most human tumors [38,39,40]. The deletion of *Atg7* in *Braf^V600E/+^-*driven lung cancer mice resulted in oxidative stress, which induced the antioxidant defense protein and nuclear factor erythrocytes-2-like 2 (NRF2), which accelerated tumor cell proliferation but reduced tumor burden at later stages of tumorigenesis and impaired tumor cell viability, thus increasing the survival of mice [41].

Interestingly, deficiencies of *Atg5* or *Atg7* in KRas mutant-derived pancreatic cancer models inhibited progression to pancreatic duct adenocarcinoma (PDAC); however, the loss of autophagy without *Trp53* no longer inhibited tumor progression but rather accelerated it. In the absence of *Trp53*, a metabolic analysis revealed that increased glucose uptake and concentration of anabolic pathways promoted tumor growth [42]. A role for autophagy in gene-specific metabolisms, such as lipid accumulation, resulting from the absence of *Trp53* in *LSL-Kras^G12D^*-derived lung cancer, was also described [35]. A deficiency of Atg7 in *Braf^V600E/+^* and *Pten^+/^^Δ^; Braf^V600E/+^*-driven melanoma mice decreased tumor growth and increased survival [43].

Multiple studies using GEM models of cancer have provided evidence to support the cancer-promoting function of autophagy in different oncogene-driven mouse cancer models (Table 1). Although autophagy depletion and genetic deletion of ATG5 and ATG7 initiate tumors in mice, these interventions ultimately slow the progression of malignant tumors in tissue-specific cancer models derived from various oncogenes [41,42,44,45].

Systemic *Atg7* ablation in *FSF-KRas^G12D^*; *Trp53 ^frt/frt^*-driven lung cancer mice increased their susceptibility to infection and neurodegeneration, thus limiting their survival up to 2–3 months; however, tumor growth decreased before the destruction of normal tissues [47]. Another group generated transgenic mice harboring *Atg4B^C74A^* with a doxycycline-inducible system and crossed them with a previously established pancreatic cancer GEM model: *LSL-KRas^G12D^, p53^lox/+^,* and *p48-Cre*. Compared with the control mice, the *Atg4B^CA^* + pancreatic cancer GEM mice exhibited slower tumor growth and increased tumor-specific survival, which was associated with autophagy inactivation [48].

Studies using tumor models revealed that autophagy causes cell death in early tumors but increases the proliferation rate during tumor development. Various mechanisms have been proposed based on tumor conditions or the microenvironment, but most involve Atg5 or Atg7. ULK1 promotes autophagy by phosphorylating several ATG protein targets, including multiple subunits. For the majority of cell lines, the absence of ULK1 alone is enough to interfere with autophagy. It has been demonstrated that the downregulation of ULK1 is sufficient to inhibit autophagy [20]. An E3 ubiquitin ligase, NEDD4L, inhibits ULK1 in pancreatic cancer cells by deregulating its protein stability and reduces tumor cell growth by exhibiting the opposite effects between ULK1 expression in the KPC mouse model and the xenograft mouse model of shNEDD4L cells [50]. Systemic KO of Ulk1 prevented liver cancer and reduced hepatic tumor growth in a diethylnitrosamine (DEN)-induced hepatocellular carcinoma (HCC) model, suggesting that ULK1 may be a target for treating liver cancer [51]. These results suggest that autophagy has a cancer cell-autonomous role during tumor progression; however, the necessity of autophagy varies among cancer type, stages, and tumor microenvironment (TME), which are discussed later in this review. Based on the results from numerous studies of molecular mechanisms, autophagy inhibition can effectively prevent advanced cancer or improve treatment effectiveness, which further warrants the development of specific autophagy inhibitors [48].

## 2. Mitophagy

### 2.1. General Mitophagy Process

Mitochondria are dynamic organelles that play a central role in both necrotic cell death (necroptosis) and programmed cell death (apoptosis), as well as cellular metabolism and survival [52]. A type of selective autophagy, known as mitophagy, has a protective role in normal physiology by eliminating unnecessary or dysfunctional mitochondria and maintaining intracellular homeostasis [53,54]. In response to acute tissue stress, mitophagy contributes to the preservation of mitochondrial integrity through multiple pathways that adjust to various environments, suggesting its significance in maintaining cell survival. During this selective autophagic process, damaged mitochondria are selectively targeted for lysosomal degradation. This process ensures mitochondrial quality control and quantity maintenance in diverse cell types, particularly in energy-demanding tissues such as the brain, skeletal muscle, heart, liver, and kidney. Thus, the interplay between mitophagy degradation and biogenesis is important for determining mitochondrial quantity and quality.

As evidence of mitophagy under normal physiology, mitochondria elimination during erythrocyte maturation depends on specific mitophagic adaptors/receptors [55]. Moreover, maternal inheritance of mitochondrial DNA (mtDNA) is involved in the selective elimination of paternal mitochondria during early embryonic development [56,57].

One key distinction between mitophagy and general autophagy is that mitophagy is activated by mutations or specific conditions that impair the electrochemical potential within the mitochondrial membrane. For example, mutations of the mitochondrial proteins, Fmc1 (formation of mitochondrial complex V assembly factor 1) and Mdm38 (mitochondrial distribution and morphology 38), compromise the internal structure and function of mitochondria, potentially leading to mitophagy [58,59,60]. In yeast, the decision to maintain or degrade mitochondria is controlled by the need for oxidative phosphorylation, thereby regulating the number of mitochondria based on cellular energy requirements and state [61,62]. Moreover, mitophagy tends to selectively remove mitochondria containing harmful mutations in the mitochondrial DNA (mtDNA) [63].

Accordingly, mitophagy serves as a pivotal mechanism for governing mitochondrial function, with signals and conditions shaping its activity, ultimately controlling mitochondrial numbers in response to cellular energy demands and state.

### 2.2. Mitophagy Related Autophagy Receptors: Mitophagy Machinery Based on the Type of Mitophagy

Mitophagy pathways are categorized by the presence of ubiquitination of the cargo substrates. The PINK1-Parkin mitophagy pathway, initially identified in neurodegenerative disease, operates through ubiquitin signaling. PINK1, encoded by PARK6, translocates to the mitochondrial membrane, where it is proteolytically processed and degraded. Parkin, which is part of the E3 ubiquitin ligase complex, is phosphorylated by PINK1 and targets various outer mitochondrial proteins, including voltage-dependent anion channel (VDAC-1) and Mitofusin 1 and 2 (Mfn1 and Mfn2). Consequently, the recruitment of mitophagy receptor/adaptors such as SQSTM1/p62, NBR1, NDP52, TAX1BP1, and OPTN provides signals for Parkin-mediated mitochondrial degradation [64] (Figure 1).

Distinct from ubiquitinated proteins in the mitochondria cargo, certain types of mitophagy receptors/adaptors directly interact with autophagosome-associated proteins, such as LC3 and GABARAP, for mitochondrial degradation. Multiple mitophagy receptors, such as BCL2/adenovirus E1B19kDa protein-interacting protein 3 (BNIP3), BNIP3-like (BNIP3L/NIX), and FUN14 domain containing 1 (FUNDC1), regulate mitophagy activity through a ubiquitin-independent pathway [65]. BNIP3 is expressed as an inactive monomer in the cytoplasm. In response to multiple stress signals, BNIP3 forms stable homodimers through its C-terminal transmembrane (TM) domain and becomes anchored to the outer mitochondrial membrane (OMM), thereby binding to LC3/GABARAP directly without additional adaptors or cargo protein ubiquitination [66].

The deletion of the transmembrane domain disrupts dimer formation and results in mitophagy defects, indicating the importance of BNIP3 homodimerization in mitophagy. Phosphorylation at the N-terminal domain of BNIP3 is required for its interaction with GABARAPL2 and LC3B. Specifically, ULK1-mediated phosphorylation at Ser17 enhances mitophagy [67] and further reduces proteasomal degradation, thus increasing BNIP3 stability [68].

BNIP3L/NIX acts as a selective autophagy receptor primarily localized to the mitochondria and peroxisomes to regulate the selective autophagy of these organelles. NIX shares significant similarities with BNIP3 and belongs to the Bcl-2 protein family, containing a BH3-only motif, thereby functioning as an equivalent to BNIP3 [69]. BNIP3 and BNIP3L/NIX have important roles in mitochondrial homeostasis through the formation of homodimers and heterodimers. Phospho-mutations in the NIX impair homodimer formation, which results in reduced LC3A-NIX recognition and compromised mitophagy [70]. In addition, phosphorylation at distinct residues in the LIR motif of NIX enhances its affinity and promotes mitochondrial influx into autophagic vesicles.

The physiological significance of NIX in mitophagy was initially demonstrated during erythrocyte maturation, which facilitates mitochondrial clearance for the transition from reticulocytes to mature red blood cells. Its absence results in mitochondrial clearance defects, resulting in compensatory expansion of erythroblasts, anemia, and erythrocyte marrow hyperplasia [71,72]. Mitophagy in NIX-deficient reticulocytes can be partially rescued by the high expression of BNIP3. BNIP3 and NIX are essential for the selective removal of mitochondria and completion of maturation in reticulocytes. NIX deficiency results in defective mitochondrial clearance in reticulocytes, leading to the compensatory expansion of erythroblasts, anemia, and erythrocytosis. Moreover, NIX-mediated mitophagy is required for retinal ganglion cell differentiation and reprogramming somatic cells into induced pluripotent stem cells (iPSCs). Although NIX is recognized for its irreplaceable role in various cellular processes, its precise regulation and activation in these processes remain to be defined [73,74].

BNIP3 or BNIP3L/NIX are rarely involved in the removal of depolarized mitochondria, which are predominantly eliminated by the Parkin/PINK systems; however, BNIP3 or NIX enhances Parkin-mediated mitophagy and further compensates for the ablation of Parkin. In addition, BNIP3 interacts with PINK1 and inhibits the proteolytic cleavage of PINK1 kinase, which results in PINK1 accumulation on the OMM, facilitates Parkin recruitment to the mitochondria, and activates mitophagy [75]. NIX can also complement mitophagic defects in the Parkin pathway. Following ubiquitination by Parkin, NIX recruits the autophagy receptor NBR1 to the mitochondria to promote mitophagy [76] (Figure 1).

FUNDC1 also serves as a direct autophagy adaptor for mitophagy under hypoxic conditions through LC3 binding [77,78]. Its activity is finely tuned by phosphorylation and dephosphorylation. ULK1 phosphorylates FUNDC1, a critical step for its recruitment to damaged mitochondria and mitophagy [79]. Under hypoxic conditions, casein kinase II (CK2)-phosphorylated FUNDC1 is dephosphorylated by phosphoglycerate mutase 5 (PGAM5). This enhances its interaction with mitochondrial fission genes, Drp1 and LC3, to form autophagosomes for damaged mitochondria elimination [80].

Overall, mitophagy is regulated by Parkin-dependent and -independent pathways, both of which are crucial for regulating mitochondrial dynamics and mitophagy (Figure 1).

### 2.3. Molecular Connections between Mitochondrial Dynamics and Mitophagy

Mitochondria are highly dynamic organelles that move within the cell and frequently undergo a division status, fission, and a merging state, fusion. In cells, a significant portion of mitochondria are interconnected within a network. The regulation of mitochondrial dynamics and mitophagy are closely intertwined. For a mitochondrion to be engulfed by an autophagosome in a prerequisite manner, it must be separated from the mitochondrial network. Fission and fusion, along with mitochondrial trafficking along actin filaments or the microtubule cytoskeleton, allow the cells to adapt to mitochondrial distribution and change local demands. Mitochondrial fission and fusion are orchestrated by a family of conserved large GTPases [81].

One such GTPase, dynamin-related protein 1 (Drp1) in mammals, has an important role in fission. Drp1 forms a multimeric complex that encircles the outer membrane of the mitochondrial tubules, applying mechanical force to induce membrane division. In contrast, mitochondrial fusion involves two distinct types of machinery, Mfn1 and Mfn2, which, in mammals, facilitate outer membrane fusion, whereas Optic atrophy 1 (Opa1) mediates inner membrane fusion. Mitochondrial fusion is also influenced by mitochondrial motility [82].

A major role of mitochondrial fusion and fission is the efficient distribution of mitochondrial DNA (mtDNA) and proteins throughout the mitochondrial network. Following fusion, mitochondria with an intact membrane potential continue to engage in fusion and fission with the mitochondrial network, whereas the depolarized daughter mitochondria are often targeted for degradation through mitophagy without undergoing the fusion stage. Mitochondrial depolarization decreases the levels of proteins, such as Opa1, Mfn1, and Mfn2, which are important for fusion, to facilitate mitochondrial fragmentation [83,84].

The importance of fission in mitophagy is supported by genetic studies. Similar to the essential role of Dnm1 (the yeast homolog of Drp1) for mitophagy in yeast, excessive fusion, induced by overexpressing Opa1 or using an inactive Drp1 mutant, prevents the autophagic degradation of mitochondria in mammals. This resistance to mitophagy due to hyperfusion also occurs under certain physiological conditions, like starvation, where a fused mitochondrial network protects mitochondria from degradation. These results highlight the need to balance mitochondrial dynamics to promote efficient mitophagy [85].

Because mitochondria act as a highly dynamic network, rather than discrete organelles, damaged mitochondria undergo a dynamic fission and fusion cycle as well as mitophagy for removal from a healthy network [86]. In this section, we provide a brief overview of how the molecular machinery involved in mitochondrial dynamics interacts with the mitophagic pathway.

To determine the functional interactions between mitochondrial dynamics and mitophagy, the involvement of the PINK1/Parkin system has been demonstrated. In fact, Mfn2 and Miro, which are components of the mitochondrial motor/adapter complex, are regulated by PINK1 and Parkin [87,88]. In mammals, Mfn2 promotes mitochondrial fusion and ER-mitochondrial tethering [89], in which the activities are regulated by PINK1-mediated phosphorylation [90]. Furthermore, PINK1-mediated phosphorylation of Mfn2 at distinct sites enables the binding of Parkin to Mfn2 and the blockade of Mfn2-mediated mitochondrial fusion, and ultimately, induces mitochondrial elimination by activating mitophagy [88,91]. The Parkin-mediated ubiquitination of Mfn1/2 induces proteasomal degradation, thereby disrupting mitochondrial fusion, separating mitochondrial-ER contact sites, and ultimately promoting mitophagy [92,93].

In addition to regulating MFN, PINK1 has a unique role in activating mitophagy, which determines the localization of Beclin1 at the mitochondrial-associated membrane (MAM) at regions of contact between the endoplasmic reticulum (ER) and mitochondria, where the autophagosomes are initiated. This function ultimately regulates the Parkin- mediated ubiquitination of substrates, its recruitment to the OMM, and mitochondrial turnover rate [94]. Moreover, PINK1 regulates mitophagy activity by phosphorylating Miro, another mitochondrial substrate at multiple sites, which also promotes its ubiquitination by Parkin and subsequent proteasomal degradation. Thus, Miro degradation and Mfn2 inhibition preferentially block mitochondrial fusion and regulate mitochondrial dynamics to facilitate mitochondrial removal by the autophagic machinery [95].

Furthermore, Parkin-independent mitophagy adaptor proteins, BNIP3 and FUNDC1 are functionally linked to the proteins involving mitochondria dynamics, Opa1 and Drp1, regulating mitophagy. BNIP3 alone inhibits mitochondrial fusion by regulating Opa1 and also promotes the mitochondrial translocation of Drp1 to lead mitochondrial fission [96,97]. Independently, FUNDC1 interacts with the proteins involving mitochondria dynamics, such as Drp1 and Opa1, facilitating mitochondrial fission and further promoting mitophagy activity [98].

### 2.4. Molecular Function of Mitophagy Regulators and Receptors/Adaptors in Cancer

Mitochondria have a profound impact on cancer etiology, progression, invasion, and metastasis. There is a significant correlation between the fate of mitochondria and cancer [99]. Because of multiple mutations in key enzymes of the tricarboxylic acid cycle (TCA) and mitochondrial oxidative phosphorylation (OXPHOS), cancer cells tend to undergo rapid metabolic reprogramming, which restricts the TCA cycle and mitochondrial OXPHOS [100]. This enables cancer cells to shift energy metabolism toward glycolysis for ATP production; however, the association between mitochondrial dysfunction and cancer is not limited to just altered metabolism [101]. Evidence linking mitochondrial dysfunction to cancer development is growing, and the connection largely depends on the cancer type. Reduced mitochondrial efficiency results in the accumulation of damaged mitochondria, which disrupts the redox balance and increases oxidative damage, including DNA mutations from reactive oxygen species (ROS). The increased production of free radicals contributes to tumor formation because of genetic instability [102]. In addition, cancer cells adapt their metabolism to reduce the regulation of ROS generated by impaired mitochondria. Accordingly, mitophagy, a selective degradation process for damaged mitochondria, is closely associated with alterations in cancer cell metabolism and further influences cancer progression.

Similar to the role of autophagy in cancer growth and survival, the role of mitophagy in cancer development is complex and depends on other factors, such as cancer type and stage. Initially, mitophagy is associated with tumor suppression during the early stages of oncogenesis by reducing ROS and genome instability. However, at later stages, mitophagy promotes tumor growth, supporting the metabolic requirements of cancer cells and resisting apoptosis, which ultimately promotes tumor development. Dysregulation of mitophagy receptors and/or regulators in cancer varies depending on the subtype and tumor microenvironment [103].

One of the canonical mitophagy regulators, the Parkin–PINK pathway, plays a tumor-suppressive role by eliminating damaged mitochondria. Loss or functional mutations in the Parkin gene have been observed in ovarian cancer, breast cancer, bladder cancer, and lung cancer [104]. Moreover, in mice lacking Parkin, liver tumors develop [105]. Mutations in the PARK6 gene (PINK1) have been observed in neuroendocrine tumors [106], suggesting that changes in mitophagy contribute to the development of specific tumor types. The autophagy regulator, AMBRA1, is an important Parkin-interacting protein in the final stages of Parkin-induced mitophagy [107]. In mice, Ambra1 is associated with tumor formation and acts as a tumor suppressor [108]. The precise role of AMBRA1 in cancer development remains unclear; however, it is clear that disruptions in AMBRA1 regulation in mitophagy are associated with carcinogenesis.

Mitophagy is typically induced by mild oxidative stress through the transcription factor HIF-1α, which is activated under hypoxic conditions. This, in turn, acts as a tumor-suppressive factor by regulating the expression of BNIP3 and NIX [109]. In contrast, some tumors utilize mitophagy as an adaptive mechanism during solid tumor formation to evade apoptosis or when cancer cells are exposed to a typical hypoxic microenvironment [110]. In fact, K-Ras-induced lung tumors require mitophagy, and its inhibition results in growth arrest, thus negatively altering the fate of the tumor [111].

VDAC1 is a mitochondrial protein that contributes to the metabolic phenotype of cancer cells by regulating mitochondrial activity [112]. VDAC1 is overexpressed in many cancer types, and silencing VDAC1 inhibits tumor development [113]. Indeed, VDAC1 serves as a mitochondrial target for Parkin and is required for the efficient targeting of damaged mitochondria [114], thus making it essential for Pink1/Parkin-mediated mitophagy [64].

This highlights the role of various other OMM proteins, beyond fusion and fission proteins, in the intricate interplay between mitochondrial fate and cancer.

Interestingly, the ablation of Parkin-independent mitophagy receptors BNIP3 or BNIP3L/NIX exhibits the opposite roles for tumor progression in different spontaneous mouse cancer models. The mitophagy receptor BNIP3 exhibits tumor-suppressive effects and significantly inhibits the growth of primary breast tumors in mouse models. This is achieved by mitigating dysfunctional mitochondria accumulation and excessive ROS production. In the absence of BNIP3, breast cancer cells are unable to eliminate dysfunctional mitochondria, which results in increased mitochondrial ROS and the upregulation of HIF-1α, a key transcription factor in tumor formation [115].

Conversely, BNIP3L/NIX, another mitophagy receptor, stimulates mitophagy in cancer cells, facilitates metabolic demands, and inhibits cell death, thus promoting tumor progression. Oncogenic KRas induces BNIP3L/NIX expression, highlighting its role in mitophagy induction and PDAC progression; however, direct evidence for mitophagy activity and mitochondrial levels during tumor progression remains limited [116].

The regulators involving mitochondria dynamics also affect tumor progression. KRas activation stimulates the activity of Drp1 to enhance mitochondrial fission. The depletion of Drp1 in KRas-induced pancreatic cancer models reduces mitochondrial activity and restrains tumor progression. Overexpressing Mfn2, which generally triggers mitochondrial fusion, is associated with decreased tumor development and enhanced survival in preclinical models, underscoring the connection between abnormally fused mitochondria and mitophagy-mediated tumor suppression [117].

The interplay between mitophagy and cancer remains complex and multifaceted. On one hand, mitophagy plays a crucial role in preventing the accumulation of damaged mitochondria that contribute to tumor formation. On the other hand, the dysregulation of mitophagy is associated with cancer cell survival, drug resistance, and evasion of cell death; however, most studies have not provided direct evidence of mitophagy during tumor progression. These processes should be examined more carefully with respect to the consequences resulting from general mitophagy defects or the identification of specific factors (Table 2).

## 3. The Role of Autophagy in Tumor Host–Cell Interactions

Although early research suggested the impacts of autophagy on the cancer cell-autonomous mechanism using various cancer mouse models, recently, more works have proposed that autophagy in the host normal cells adjacent to the tumor also plays a critical role in cancer progression, which contributes to various tumor microenvironment cells adjacent to tumors, such as cancer-associated stromal cells and tumor-associated immune cells [127] (Figure 3).

### 3.1. Host Autophagy Promotes Tumor Cell Metabolic Deficiencies

Tumors arise through interactions with the host stromal and immune cells. Specifically, in orthotopic models of PDAC, autophagy in pancreatic stellate cells (PSCs) has an important role in generating and secreting nonessential amino acids, such as alanine, extracellularly to promote the growth of pancreatic cancer cells [128]. In addition, the loss of hepatic- and host-specific autophagy results in the release of the arginine-degrading enzyme arginase 1 (ARG1) into the bloodstream, resulting in a decrease in circulating arginine levels and rendering the primary tumor unable to sustain growth. Dietary arginine supplementation in Atg7-deficient hosts partially restores circulating arginine levels and tumor growth, thus revealing a novel metabolic vulnerability in cancer from an autophagic defect in the host. This causes the tumor to have a nutritional requirement for arginine, potentially rendering it a promising target for autophagy inhibition in the liver [129].

This was further examined using an autophagy inhibition model achieved through the inducible expression of a dominant-negative Atg4B^CA^ in KRas mutant-driven pancreatic cancer GEM model to co-express Atg4B^CA^ with mutants Kras and Trp53 in all pancreatic cells under the control with doxycycline. Atg4B^CA^ sequesters free LC3B, preventing its lipidation, and consequently impairs autophagosome maturation. In this model, Atg4B^CA^-mediated autophagy inhibition in Kras-driven pancreatic tumors resulted in tumor regression. Moreover, to observe the effect of whole-body Atg4B^CA^ expression on tumor growth, the orthotopic implantation of Kras-driven pancreatic cancer cells expressing or lacking Atg4B dominant negative (DN) in hosts, with or without systemic Atg4B^CA^ expression, revealed that stromal cell ATG4B enhanced early tumor establishment, but its role diminished over time, with Atg4B DN in tumor cells becoming more significant in slowing tumor growth. This study highlights the distinct contribution of host and tumor cell autophagy to tumor growth, emphasizing the prominent role of surrounding cells in tumor implantation and the minor role of supporting established tumor growth. It also revealed that autophagy inhibition in both host and tumor cells, through intricate combinations for in vivo tumor assessment and autophagy measurement, impacts tumor growth. Furthermore, it also highlights the distinct contributions of host and tumor cell autophagy to tumor growth, emphasizing the prominent role of surrounding cells in tumor transplantation and a key role in promoting tumor development [48].

More recently, systemic autophagy inhibition through Atg5 knockdown significantly reduced established tumor growth, which is associated with defective glucose and lactate uptake in Kras-driven lung cancer mice. The systemic autophagy loss by transient Atg5 deletion metabolically reduced the carbon source from glucose and lactate in *Kras^G12D/+^; p53^−/−^* (KP)-driven lung tumors; thus, impaired tumor growth was observed in these mice. Interestingly systemic Atg5 depletion also promoted the infiltration of cytotoxic T cells toward tumors, enhancing the anti-tumor effect and increasing the survival rate of these lung tumor-harboring mice. This study raises a new possibility for how host cell autophagy can have a more widespread impact on metabolite exchange between the host and tumor, highlighting its potential significance in the context of the interplay between host and tumor cells [130].

Overall, these studies underscore the important role of autophagy in diverse host cells, with a focus on the provision of essential metabolites, particularly specific amino acids vital for fueling the metabolism of proliferating tumors. Furthermore, they propose targeting systemic therapeutic autophagy as an anti-cancer strategy, thus surpassing tumor cell-specific autophagy inhibition.

### 3.2. Autophagy Maintains the Activity of Cancer-Associated Fibroblasts (CAFs)

Among diverse host cells, a key role of stromal cell autophagy is closely linked to tumor formation. The role of stromal cell autophagy has primarily been determined through research on cancer-associated fibroblasts (CAFs), which regulate cancer proliferation and are present in the majority of solid tumors. CAFs are a key component of the tumor stroma and play an important role in mediating crosstalk between cancer cells and the tumor microenvironment. Interestingly, CAFs are also more resistant to stress compared with normal fibroblasts, showing recovery by Beclin-1 or Atg5 knockdown. This suggests that autophagy directly affects CAFs during cancer development [131].

CAFs exhibit increased autophagy and impaired mitochondrial function, resulting in the secretion of metabolites, such as glutamine or alanine, which are subsequently taken up and utilized by cancer cells. Glutamine in cancer cells drives autophagy in stromal cells by inducing ammonia production and increasing the mitochondrial TCA cycle and oxidative phosphorylation. This phenotype results in increased ATP production and further enhances the anabolic metabolism for effectively producing daughter cells.

Breast cancer type 1 susceptibility protein (BRCA1) and caveolin-1 (Cav-1) are involved in the oxidative stress pathways of CAFs. Stromal-specific BRCA1 depletion causes tumor growth by inducing HIF-1a levels, increasing the ketone body, and activating autophagy/mitophagy [132]. In a co-culture system of breast cancer cells and CAFs, adjacent CAFs can induce oxidative stress, which promotes cancer progression by reducing the levels of stromal Cav-1 and increasing the expression of autophagy markers [133]. Cav-1 knockdown in fibroblasts appears to protect breast cancer cell death in co-culture, thus indicating another mechanism through which CAFs can promote cancer cell survival [134]. Accordingly, low levels of stromal Cav-1 are associated with poor prognosis in various cancers, including breast [135], prostate [136], and colorectal cancer [137].

Because the role of autophagy in stromal cells has been emphasized in the supply of important metabolites that can compensate for the metabolic defects in highly proliferating cancer cells for growth and survival, early studies indicated that CAF cells with a loss of Cav-1 enhance autophagy, which may play a role in mitochondrial energy generation in tumor cells [138]. In a co-culture system of breast cancer cells and non-transformed fibroblasts, glutamine treatment increased mitochondrial mass, which was abolished with an autophagy inhibitor. Accordingly, glutamine enhanced the levels of autophagy markers in fibroblasts, suggesting that autophagy-activated fibroblasts may provide glutamine to support mitochondrial metabolic activity in cancer cells [139].

Autophagic alanine secretion by pancreatic stellate cells (PSCs) may be a key mechanism in driving tumor growth. This secreted alanine is also utilized as a replacement carbon source derived from glucose and glutamine in PDAC, which fuels the tricarboxylic acid (TCA) cycle, nonessential amino acid (NEAA) synthesis, and lipid biosynthesis [128]. The maintenance of metabolic crosstalk between tumors and adjacent stromal cells relies on autophagic processes, which generate alanine from PSCs as a pivotal carbon source, thereby facilitating the metabolic reprogramming of cancer cells (Figure 1).

Interestingly, nucleosides can also be secreted by CAFs, which have an important role in PDAC growth through autophagy. As a novel autophagy regulator, nuclear fragile X mental retardation-interacting protein 1 (NUFIP1) activates autophagy for secreting nucleosides from CAFs. These CAF-derived nucleosides induce glucose consumption and further promote PDAC proliferation. The suppression of nucleoside secretion by targeting NUFIP1 in CAFs significantly reduced tumor weight in a PDAC mouse model [140].

One of the major roles of autophagy in CAFs is the active secretion of various intracellular proteins such as growth factors, inflammatory cytokines, extracellular matrix components, and even exosomes [141]. Increased autophagy in stromal fibroblasts was correlated with adverse patient outcomes in head and neck cancers and inhibiting autophagy in fibroblasts was associated with reduced tumor progression in an in vitro co-culture model because of the reduced secretion of various protumorigenic factors, including IL-6, IL-8, and fibroblast growth factors (FGFs) [142]. CAF autophagy is also linked to secretory events required for the formation of a desmoplastic stromal response. Autophagy in PSCs, which are responsible for generating the fibrotic stromal matrix observed in PDAC, promotes the secretion of extracellular matrix (ECM) components and inflammatory cytokines in CAFs [143].

A genetic deficiency of autophagy in CAFs in both spontaneous and orthotopic transplantation breast tumor models driven by the PyMT oncogene is sufficient to inhibit tumor growth and improve tumor-related survival. Specifically, the genetic loss of autophagy in stellate cells induces a particular defect in collagen protein-degrading enzymes and impairs the secretion of type I collagen in vitro and in vivo. In addition to its effects on type I collagen secretion and tissue stiffness, a deficiency of autophagy in stellate cells supports the role of stellate cell autophagy in regulating tumors through multiple secretory pathways by reducing the secretion of inflammatory cytokines and angiogenic factors [144,145]. Moreover, autophagy deficiency in CAFs inhibits CAF activation through defective proline biosynthesis and the mitophagy-mediated regulation of NADK2 (NAD kinase 2), an enzyme required for the production of mitochondrial NADP(H). Indeed, the suppression of Parkin-mediated mitophagy in CAFs significantly reduced tumor size in a PDAC mouse model. Accordingly, targeting autophagy/mitophagy in CAFs may represent a promising potential anti-cancer treatment strategy [146].

In addition to tumor cells, exosomes are also secreted by CAFs. Exosomes secreted by patient-derived CAFs reprogram the metabolic machinery following uptake by cancer cells. CAF-derived exosomes provide exosomal cargo and disrupt mitochondrial oxidative metabolism. CAF-derived exosomes increase glutamine reductive carboxylation for macromolecule synthesis in prostate cancer cells [147].

Under hypoxic conditions, CAFs and tumor cells secrete a significant number of exosomes, which are regulated by autophagy and support the metabolic demand of cancer cells. Under hypoxic conditions, tumor cells tend to secrete proangiogenic factors that are involved in blood vessel generation and the restoration of an oxygen supply to tumor cells. Thus, exosomes, which are also known as extracellular vesicles, are mediators of cell-to-cell communication within the tumor microenvironment.

A recent study suggested that GABARAPL1, a member of the LC3/GABARAP protein family, plays a crucial role in endosomal maturation and exosome secretion and promotes the loading of cargo and the generation of pro-angiogenic exosomes in hypoxic tumor cells [148]. Hypoxia-activated ATM kinase directly phosphorylates BNIP3, a mitophagy receptor, to stimulate autophagosome formation and exosome release from hypoxic breast CAFs. The genetic deletion of ATM or BNIP3 suppresses autophagy activity and exosome secretion from hypoxic CAFs (Figure 1). Moreover, GPR64, which is enriched in CAF-derived exosomes, stimulates non-canonical NF-κB to upregulate MMP9 and IL-8 in breast cancer cells and enhance the invasive phenotypes of cancer cells. This study suggests a novel regulatory mechanism of CAFs in promoting tumor progression, in which oxidized ATM stimulates autophagy and autophagy-mediated exosome release [149]. In addition, integrin beta 4 (ITGB4)-overexpressing triple-negative breast cancer (TNBC) cells provide the cancer-promoting capabilities of CAFs with ITGB4 proteins via exosomes, thus enhancing BNIP3L-dependent mitophagy and producing more lactate in CAFs [150].

Overall, these studies emphasize the central role of stromal autophagy in primary tumor progression and elucidate working mechanisms that contribute to the potential effect of autophagy inhibition as an anti-cancer strategy (Figure 3).

### 3.3. Autophagy and the Immune Response against Tumors

Early studies indicated that lysosomal degradation products are provided to CD4+ T cells through major histocompatibility complex (MHC) class II molecules and coordinate the specific immune response. Autophagy can regulate the effect of stimulating the adaptive immune response by interacting with antigen-processing pathways in dendritic cells [151]. Autophagy is implicated in cytoplasmic and viral antigen presentation on MHC class II molecules. To address the role of autophagy in antigen presentation of tumors, apoptotic tumor cells were administered to mice with dendritic cell-specific deletion of Atg5. The mice exhibited a reduction in the response of CD4+ T cells but not CD8+ cytotoxic T cells, which results in the inhibition of MHC-II antigen presentation [152].

Previous studies have indicated a positive role for autophagy in antigen presentation. Autophagy mediates antigen generation by cargo degradation, and these antigens are subsequently presented on the cell surface for recognition by immune cells [153,154]. Moreover, numerous reports suggest that autophagy is required for the differentiation of hematopoietic cells [155] and maintaining homeostasis of systemic immunity, such as supporting the functional integrity of memory and effector T cells [156].

In contrast, recent reports have suggested the opposite evidence, stating that autophagy inhibition boosts the immune system to block tumor progression. In general, autophagy acts as a cell survival pathway to block cell death, particularly mediated by certain functional immune cells. Contrary to the supportive role of autophagy in CD4+ T cell stimulation, autophagy was shown to block antigen presentation to MHC-I molecules by enhancing MHC class I internalization in dendritic cells (DCs) [157]. As opposed to viral antigen presentation on MHC-II mediated by autophagy, MHC-I expression increases in dendritic cells in the absence of Atg5 and Atg7, which results from the defect in autophagy degradation, indicating that autophagy can also degrade MHC-I in dendritic cells [158]. Although the evidence supporting that autophagy inhibition facilitates immune activation for suppressing tumors is different from the autophagy-mediated antigen-presentation function in normal physiology, a series of studies have suggested that autophagy inhibition in various cancer models revealed the synergistic effect with immune checkpoint inhibitors (ICIs) anti-cancer therapy through multiple mechanisms.

Accordingly, highly active autophagy in pancreatic cancer plays an important role in the immune evasion of cancer by the selective elimination of MHC-I [159]. As a selective autophagy adaptor, NBR1-mediated MHC-I degradation enables the cytotoxic efficacy of natural killer (NK) or effector T cells to attack tumors, and thus autophagy inhibition restores MHC-1 levels, thereby enhancing antigen-presenting capability and T cell immunity in a pancreatic cancer model. Based on this immune-modulating feature of targeting autophagy, this study suggests that autophagy inhibition provides a synergistic anti-cancer effect with immune checkpoint inhibitors (ICIs) (Figure 1).

In an early study, the genetic deletion of autophagy-related genes blocks tumor progression in several mouse tumor models. This inhibition boosts the release of various chemokines from tumors, ultimately facilitating tumor-suppressive immunity. For example, in the PyMT mammary tumor mouse model, Fip200 deletion blocks cancer progression by enhancing chemokine production such, as CXCL9 and CXCL10, and further recruiting CD8+ cytotoxic T cells into tumors [46] (Figure 1). In addition, the deletion of another autophagy regulator, Beclin1, inhibited tumor growth in melanoma models by enhancing natural killer (NK) cell infiltration into the tumor region, and the anti-tumor effect is mechanistically regulated by the increased levels of CCL5 chemokine. The high expression of CCL5 is also shown in melanoma patients with higher survival rates [160]. Moreover, the targeting of Vps34, a class III PI3 lipid kinase, in some cancer cells suppresses cancer growth, mediated by infiltrating NK and cytotoxic T cells. These anti-tumor immune cell infiltrations by Vps34 inhibition are caused by the upregulation of certain chemokines, CCL5 and CXCL10, thereby improving ICI therapeutic efficacy in melanoma and colorectal cancer mouse models [161].

Another report suggested that because LKB1-deficient lung cancer exhibiting high autophagy activity tends to decrease MHC presentation through autophagic degradation, autophagy inhibition enables the recovery of MHC class antigen presentation and further enhances the therapeutic effect of immune checkpoint inhibitors (ICIs) in lung cancer models [162]. In liver cancer, autophagy inhibition suppresses tumor progression by promoting anti-tumor effector T cell responses, which are particularly dependent on the mutational load in tumors. This study suggests that autophagy suppresses inflammation and IFN type I and II responses, which facilitates the growth of cancers with a high-tumor mutational burden (TMB), such as liver cancer [163].

Recently, ULK1 has been identified as an effective target of IFNγ signaling, which is positively associated with resistance to ICI therapy. Upregulated ULK1 in melanoma cells is positively correlated with the IFNγ-induced expression of immunosuppressive genes, such as PD-L1 and PD-L2. The inhibition of ULK1 substantially reduces IFNγ-induced PD-L1 and PD-L2 expression in melanoma cells. Accordingly, the combination treatment of a ULK1 inhibitor with anti-PD1 increases tumor regression in a subcutaneous melanoma implant model through a reduction in PD-L1/L2 levels and enhances the infiltrating efficacy of anti-tumor immune cells [164]. In addition, ATG16L1, highly expressed in kRas-mutant tumors showing poor clinical response to anti-PD-L1 therapy, has been suggested as a critical autophagy target for regulating anti-tumorigenic immunity. The depletion of the Atg16l1 gene in mouse colon cancer organoids inhibited tumor growth in the syngeneic mice model by increasing sensitivity to IFN-γ-mediated responses. These results suggest that autophagy is actively involved in immune evasion, further suggesting the synergistic effect of an autophagy blockade in combination with anti-cancer immunotherapy [49]. Moreover, lysosomal inhibition by targeting palmitoyl-protein thioesterase 1 (PPT1) pharmacologically resulted in effective cell death mediated by lysosomal lipid peroxidation, leading to regulating anti-tumorigenic T cell activities [165].

More directly, autophagy maintains the functional integrity of immune-suppressive regulatory T cells (T reg), thereby facilitating tumor progression by activated T reg cells. Thus, T reg cells utilize autophagy to maintain the immunosuppressive effect to kill tumors more effectively along with activated cytotoxic T cells and NK cells [166]. In addition, T cells depleted of Atg5 or Atg7 exhibit decreased tumor progression in syngeneic mouse tumor models resulting from the enhancement of glucose metabolism in the effector T cells, which suggests that the cell-autonomous effect for autophagy in T cells contributes to tumor-suppressing activity [167]. Autophagy also contributes to immune surveillance clearance in nascent tumor cells. In an oncogenic kRas-driven lung cancer mouse model, Atg5 deficiency is associated with an increase in the numbers of early hyperplastic foci and regulatory T cells, and then further antibody treatment or depletion of FoxP3+ cells decrease the hyperplastic lesions to those seen in the controls [36].

### 3.4. Mitophagy in the Cancer Microenvironment

In addition to autophagy, the regulation of mitochondria integrity and degradation within the tumor microenvironment plays an important role in cancer progression. Fibroblasts adjacent to tumors regulate autophagy and mitophagy to support the metabolic demands of cancer cell growth, proliferation, migration, and invasion [168,169].

A couple of studies propose that autophagy/mitophagy and aerobic glycolysis processes indeed converge in the tumor microenvironment, named “2-compartment tumor metabolism” or “parasitic cancer metabolism” model [170,171]. Based on this model, reactive oxygen species (ROS) generated by cancer cells are transferred to neighboring CAFs or other stromal cells to initiate mitophagy and oxidative stress responses. Subsequently, mitochondrial dysfunction in stromal cells results in the production of high-energy metabolites, such as L-lactate, ketones, glutamine, and free fatty acids, which promote cancer cell survival. This “host-parasite” relationship that existed between tumor stromal cells and epithelial cancer cells is known as the “re-verse Warburg effect”. Earlier studies indicated that the loss of stromal CAV1, which can be induced by ROS emitted from cancer cells, eventually results in stromal CAV1 loss [135,172].

Mitophagy and mitochondrial dysfunction reprogram stromal cell metabolism within the tumor microenvironment. In particular, the upregulation of two key cancer-related microRNAs, MIR31 and MIR34C, is sufficient to induce mitophagy, as they are closely associated with known mitophagy-inducing factors, such as oxidative stress and hypoxia-inducible factor 1A (HIF1A) activation. These results provide new insights into the role of mitophagy in tumor formation in the context of microRNA regulation [173].

CAV1 can negatively regulate transforming growth factor-beta (TGF-β) signaling; however, the activation of TGF-β signaling in tumor stromal cells appears to be necessary to induce mitophagy. In fact, paracrine or autocrine activation of TGF-β signaling in stromal cells, rather than ligand-dependent cancer cells, induces metabolic reprogramming of the tumor microenvironment [174]. In addition, hydrogen peroxide generated by ovarian cancer cells lacking BRCA1 also induces mitophagy by activating the nuclear factor kappa-light-chain-enhancer of activated B cell (NFκB) signaling, which is known for suppressing BRCA1, a tumor suppressor gene that maintains genome integrity and suppresses tumors by activating NFκB signaling in stromal cells. However, cancer cells lacking BRCA1 affect the metabolic reprogramming of neighboring CAFs by activating NFκB signaling, thereby inducing mitophagy [175].

In addition to ROS, cytokines or physiological factors enriched in the tumor microenvironment, such as the migration stimulating factor (MSF), a genetically truncated N-terminal isoform of fibronectin, can also induce mitophagy in tumor stromal cells by activating TGF-β and CDC42- NFκB signaling [176]. Overall, mitophagy is primarily activated through oncogenic signaling pathways, including TGF-β and NFκB, and can promote cancer cell growth by reprograming cancer cell metabolism [176]. Therefore, neutralizing ROS or metabolically isolating cancer cells from adjacent and supportive stromal cells can utilize mitophagy as an effective strategy for cancer cells. Accordingly, inactivating TGF-β and NFκB signaling in the tumor microenvironment provides valuable insights for targeting mitophagy.

## 4. Conclusions and Future Directions

Autophagy serves as a vital quality control mechanism in cellular homeostasis, safeguarding cells from metabolic stresses and influencing various physiological processes, including development, aging, and immunity. Additionally, its involvement in diverse diseases, such as neurodegenerative and metabolic disorders, underscores its significance. Mitophagy, a selective form of autophagy, also plays a pivotal role in maintaining mitochondrial integrity, with implications for normal physiological functions, like reticulocyte maturation.

Autophagy exhibits dual roles in cancer, acting as both a tumor suppressor and promoter, which suggests its potential as a promising anti-cancer target pathway. Multiple in vivo tumorigenesis studies using GEM models derived from various oncogenic and/or tumor suppressor mutations have demonstrated that autophagy inhibition effectively hinders tumor progression of various malignant tumor types. These findings support the development of autophagy-modulating drugs for cancer treatment [103,177].

Moreover, autophagy within host cells adjacent to tumors significantly contributes to cancer progression, particularly in the strategy of metabolic symbiosis between cancer and the tumor microenvironment (TME). For example, autophagy and mitophagy in CAF cells tend to secrete various nutrients, such as amino acids and nucleotides, to support tumor progression. In addition, autophagy in immune cells plays a role in tumor-promoting cytokine secretion, contributing to tumor expansion. This review emphasizes the discussion of various mechanisms through which autophagy and mitophagy in the TME impact tumor progression; however, the specific molecular contributions of autophagy to metabolic adaptation in cancer remain largely unknown, particularly given the diverse metabolic alterations driven by specific oncogenes or tumor suppressor genes. A selective autophagy, mitophagy regulation, necessitates a deep molecular understanding and scientific approaches to address the complex challenges.

Based on these findings, future studies into the intricate mechanisms of autophagy and mitophagy, especially in the context of cancer, hold great promise for the development of novel therapeutic strategies. Moreover, a deeper exploration of the roles of autophagy and mitophagy in regulating cancer immunity could also position these pathways as immune-modulating targets in cancer. A comprehensive understanding of these pathways will pave the way for innovative cancer treatments, personalized interventions, and targeted strategies within the complex and dynamic cancer microenvironment. This review underscores the importance of continued research to unveil the full potential of autophagy and mitophagy in shaping the landscape of cancer therapy.

## Figures and Tables

**Figure 1 cells-12-02742-f001:**
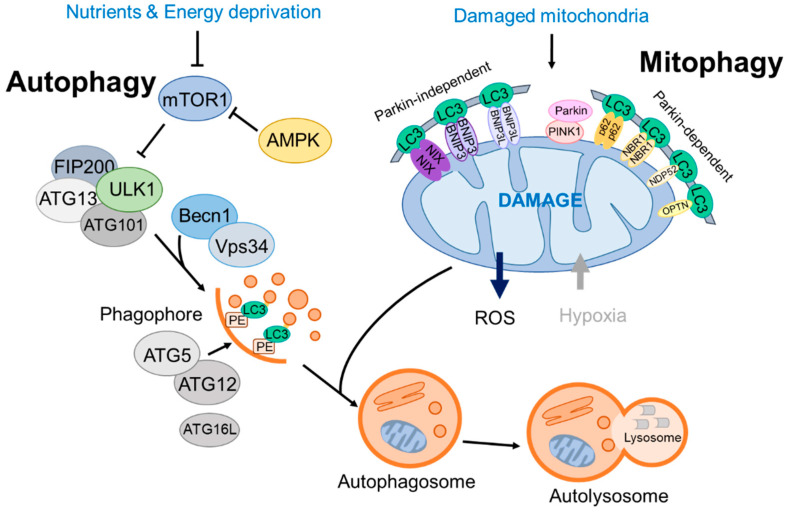
A brief overview of autophagy and mitophagy processes.

**Figure 2 cells-12-02742-f002:**
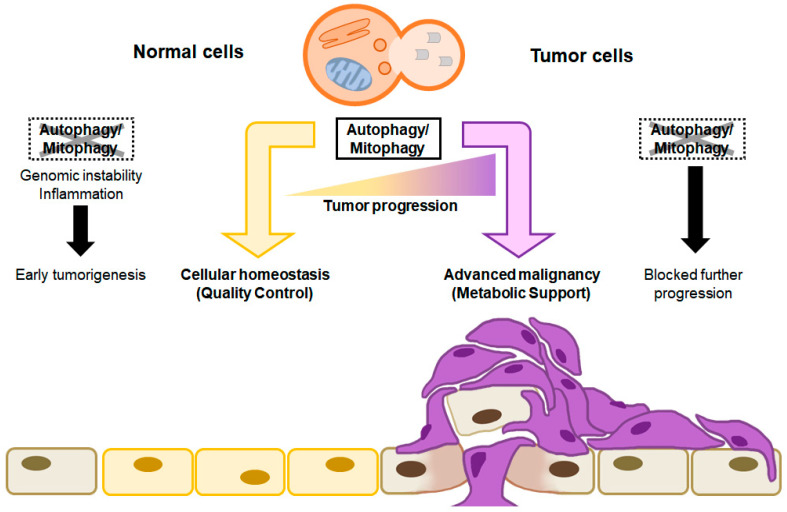
The context-dependent role of autophagy in tumor progression. Autophagy/mitophagy in normal cells maintains cellular homeostasis by eliminating malfunctioning organelles, misfolded protein, and unnecessary cytoplasmic components. When autophagy is not carried out, accumulated cellular damage, such as genomic instability and inflammation, might initiate tumorigenesis (**left**). On the other hand, autophagy/mitophagy in tumor cells promotes cell growth and survival by recycling degraded proteins and damaged organelles as energy sources, which results in advanced malignancy. When autophagy is not carried out in tumor cells, no further tumor progression occurs (**right**).

**Figure 3 cells-12-02742-f003:**
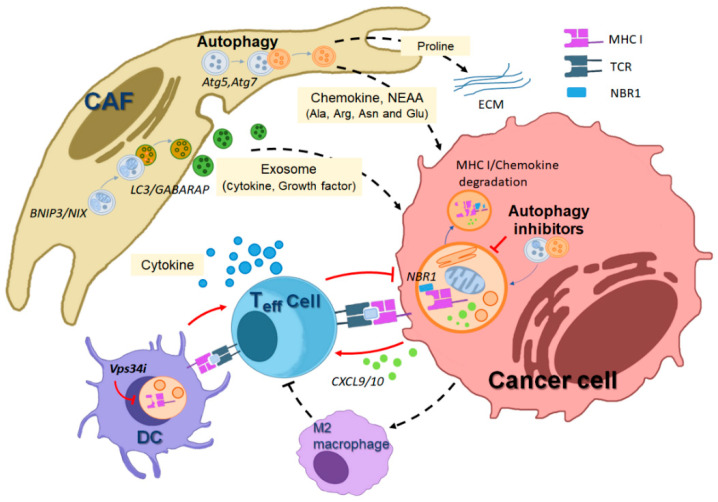
Illustration depicting how autophagy in various host cells within the tumor microenvironment (TME) mediates crosstalk that supports tumor promotion. Autophagy inhibition affects the anti-tumorigenic roles of diverse host cells in the TME. In host stromal cells, such as cancer-associated fibroblasts (CAFs), autophagy promotes the generation of cellular metabolites, including amino acids derived from macromolecule degradation, supporting tumor cell growth and survival. Additionally, CAFs generate exosomes in an LC3/GABARAP-mediated manner, which are secreted and contain pro-tumorigenic growth factors and cytokines, further promoting tumor cell proliferation. Autophagy-driven pro-collagen degradation and proline production also contribute to extracellular matrix (ECM) production, facilitating tumor proliferation. In host immune cells adjacent to tumors, autophagy plays a role in the degradation of MHC class I mediated by NBR1, an autophagy receptor, thereby inhibiting cytotoxic T cell-mediated tumor killing. Autophagy inhibition in dendritic cells (DCs) or cancer cells leads to increased MHC-I levels on the cell surface by preventing autophagic degradation, thus supporting cytotoxic T cell-mediated cancer death. The black dotted line indicates a tumor-promoting effect, while the red line represents a tumor-suppressing effect. (MHC I; major histocompatibility complex I, TCR; T cell receptor, NBR1; neighbor of the Brca1 gene).

**Table 1 cells-12-02742-t001:** The effects of absent autophagy in GEM models of cancer.

Genotype	Deletion	Effect	Type	Reference
*CAG-Cre, Alb-Cre*	*Atg5, Atg7*	Increased progression	Hepatic adenomas	[29]
*LSL-KRas^G12D^*	*Atg5*	Increased initiation, decreased progression	Lung cancer	[36]
*MMTV-PyMT*	*Fip200*	Increased initiation, decreased progression	Mammary adenocarcinomas	[46]
*TyrCre/Pten+/Δ;Braf^V600E/+^*	*Atg7*	Decreased progression	Melanoma	[43]
*LSL-KRas^G12D^*	*Atg5, Atg7*	Decreased progression with Trp53	Pancreatic cancer	[42]
*LSL-KrRas^G12D^*	*Atg7*	Decreased progression, increased life span without Trp53	Lung cancer	[35]
*FSF-KRas^G12D^; Trp53^frt/frt^; Ubc-CreERT2*	*Atg7*	Decreased progression	Lung cancer	[47]
*Braf^V600E^/+*	*Atg7*	Increased initiation, decreased progression, increased life span without Trp53	Lung cancer	[41]
*LSL-KRas^G12D^; Trp53^L/+^*	*Atg5*	Decreased progression	Pancreatic cancer	[44]
*LSL-KRas^G12D^, Trp53^lox/+^, p48Cre^+^*	*Atg4^CA^*	Decreased progression	Pancreatic cancer	[48]
*B6-129S7-IFNgtm1TS/J;*	*Atg 16l*	Decreased progression	Colorectal cancer	[49]
*K5-SOS* *EGFR^wa2/wa2^; K14-Cre*	*Atg7*	Decreased progression	Epithelial skin cancer	[37]

**Table 2 cells-12-02742-t002:** The molecular effects of mitophagy in cancer.

Target Gene	Effect	Type	Reference
Parkin	Deficiency in the expression of PARK2 is significantly associated with adenomatous polyposis coli (APC) deficiency in human colorectal cancer	Colorectal cancer (tumor suppressor)	[118]
PINK1	A high expression of PINK1 increases proliferation in NSCLC progression and chemoresistance	Lung cancer (oncogene)	[119,120]
BNIP3	The silencing of BNIP3 expression was associated with methylation of the hypoxia-responsive element (HRE) site which, in turn, inhibited the binding of HIF-1α to the BNIP3 promoter	Pancreatic cancer (tumor suppressor)	[121]
NIX	Mitochondrial NIX expression is enriched in pseudopalisading cells surrounding the hypoxic of glioblastoma and supports tumor cell survival	Brain cancer (oncogene)	[122]
	NIX knockdown significantly delays the progression of pancreatic cancer and improves survival rates in a mouse model of PDAC (pancreatic ductal adenocarcinoma)	Pancreatic cancer (oncogene)	[116]
FUNDC1	FUNDC1 expression confers the cellular and metabolic features that support cancer cell proliferation	Prostate, lung, and breast adenocarcinoma (oncogene)	[123]
PGAM5	Elevated PGAM5 expression in HCC is associated with a poor prognostic phenotype. Knocking down PGAM5 in HCC cells inhibited cell viability and enhanced chemosensitivity	Hepatocellular carcinoma (oncogene)	[124]
AMBRA1	The absence of Ambra1 promotes the formation of melanocytic nevi and accelerates melanoma growth, eventually enhancing metastatic potential	Melanoma (oncogene)	[125]
DRP1	In a KRas-induced pancreatic cancer tumor model, Drp1 plays a crucial role in both the tumorigenic process and oxidative metabolism. Inhibiting Drp1 increases the survival rate in pancreatic cancer	Pancreatic cancer (oncogene)	[126]
MFN2	The oral drug leflunomide, which is used for arthritis, promotes a twofold increase in Mfn2 expression in a pancreatic ductal adenocarcinoma (PDAC) model, leading to a 50% improvement in the average survival rate of mice with tumors compared to the vehicle	Pancreatic cancer (tumor suppressor)	[117]

## Data Availability

Not applicable.

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
