# Peer review of "Unraveling the Intricacies of Autophagy and Mitophagy: Implications in Cancer Biology"

_cells, 2023, doi:10.3390/cells12232742_

Round 1
Reviewer 1 Report
Comments and Suggestions for Authors
The article is generally well-written review on the role of autophagy and mitophagy in cancer biology by authors who have made contributions to the field. My suggestions are:
Major points:
1. There are numerous review articles on the role of autophagy and mitophagy in cancer, what is special about this paper? Perhaps it should be emphasized more that this article also deals with the role of autophagy and mitophagy in the stromal environment and the immune response.
2. Similar to the above, for the Conclusion section, it would be useful to summarize the role of autophagy and mitophagy not only in cancer cells, but also in CAFs and immune cells.
3. It might be useful to include Figures showing principal members of autophagy and mitophagy pathways
4. Lines 492 -505. this paragraph describes the image including abbreviations and references are completely missing, is that paragraph part of the Figure legend?
5. Table 2 - it would be useful to describe the method used to establish the role of NIX for ref. 122 and FUNDC1 for ref 123 as described for other targets
Minor points:
Line 133-135: the sentence is missing a verb
Line 149: liver-specific
Line 362 and line 587 – typing errors
Author Response
- There are numerous review articles on the role of autophagy and mitophagy in cancer, what is special about this paper? Perhaps it should be emphasized more that this article also deals with the role of autophagy and mitophagy in the stromal environment and the immune response.
(Answer) Thank you for the reviewer's critical point and suggestion. Following the reviewer's suggestion, we emphasize the role of autophagy and mitophagy in the stromal environment and the immune response, in conclusion as shown in the yellow mark. Moreover, we add a new figure (figure 2) to explain for dealing with the dual functions of these pathways in cancer and TME.
- Similar to the above, for the Conclusion section, it would be useful to summarize the role of autophagy and mitophagy not only in cancer cells but also in CAFs and immune cells.
(Answer) Following the reviewer's suggestion, we add more explanation about the general role of autophagy and mitophagy in normal physiology and summarize the importance of autophagy and mitophagy in tumor microenvironment., in the conclusion section as marked in yellow.
- It might be useful to include Figures showing principal members of autophagy and mitophagy pathways
(Answer) We add Figure 1 showing the principal members of autophagy and mitophagy pathways
- Lines 492 -505. this paragraph describes the image including abbreviations and references that are completely missing, is that paragraph part of the Figure legend?
(Answer) Yes, this paragraph is part of the figure legend, so we used a small size of font. In the revised manuscript, we use bold font to distinguish the text and the legend. Also following your suggestion, we put the full names for each abbreviation at the end of the legend as yellow marked.
- Table 2 - it would be useful to describe the method used to establish the role of NIX for ref. 122 and FUNDC1 for ref 123 as described for other targets
(Answer) We revise to describe the way to explain the role of NIX and FUNDC for ref 122, and 123 in Table 2, which are similar to the one for other molecules.
Minor points:
Line 133-135: the sentence is missing a verb
Line 149: liver-specific
Line 362 and line 587 – typing errors
(Answer) Thank you for pointing out the errors in the manuscript. We revised all the lines the reviewer mentioned in the revised version, as marked in yellow.
Reviewer 2 Report
Comments and Suggestions for Authors
In their review, Lee et al. describe the multifaceted roles of autophagy and mitophagy in normal physiology, as well as in cancer. The review is divided in 3 main parts: Comprehensive autophagic processes; Mitophagy; and the role of autophagy in tumor-host cell interactions. In this manuscript, the authors describe the mechanisms responsible for normal autophagy and mitophagy and the dual function as well as the molecular mechanisms of autophagy and mitophagy in cancer. They further explore how these two processes influence tumor-host cell interaction and highlight the potential of autophagy as immunomodulatory target in cancer.
While the review is very well written and the text clear, figures that would summarize the text are really lacking and would really help the reader. For example, the authors should consider making one or two figures summarizing the mechanistic processes in normal autophagy, and its dual function in cancer. Likewise, another figure or two should show the mechanisms of mitophagy in normal cells and in cancer cells. This would greatly enhance the readability of the manuscript.
Comments on the Quality of English LanguageEnglish requires very minor editing. Text is very well written.
Author Response
In their review, Lee et al. describe the multifaceted roles of autophagy and mitophagy in normal physiology, as well as in cancer. The review is divided in 3 main parts: Comprehensive autophagic processes; Mitophagy; and the role of autophagy in tumor-host cell interactions. In this manuscript, the authors describe the mechanisms responsible for normal autophagy and mitophagy and the dual function as well as the molecular mechanisms of autophagy and mitophagy in cancer. They further explore how these two processes influence tumor-host cell interaction and highlight the potential of autophagy as immunomodulatory target in cancer.
While the review is very well written and the text clear, figures that would summarize the text are really lacking and would really help the reader. For example, the authors should consider making one or two figures summarizing the mechanistic processes in normal autophagy, and its dual function in cancer. Likewise, another figure or two should show the mechanisms of mitophagy in normal cells and in cancer cells. This would greatly enhance the readability of the manuscript.
(Answer) Thank you for your constructive suggestion. Following the reviewer's suggestion, we add two more new figures summarizing the mechanistic processes in normal autophagy (figure1), and its dual function in cancer (figure2). However, the impacts of mitophagy in normal cells on near-cancer cell progression have not been fully understood yet, we just delineated the role of mitophagy receptors in different cancer types in Table 2.
Round 2
Reviewer 2 Report
Comments and Suggestions for Authors
In their review, Lee et al. describe the multifaceted roles of autophagy and mitophagy in normal physiology, as well as in cancer. The review is divided in 3 main parts: Comprehensive autophagic processes; Mitophagy; and the role of autophagy in tumor-host cell interactions. In this manuscript, the authors describe the mechanisms responsible for normal autophagy and mitophagy and the dual function as well as the molecular mechanisms of autophagy and mitophagy in cancer. They further explore how these two processes influence tumor-host cell interaction and highlight the potential of autophagy as immunomodulatory target in cancer.
Previous comments have been addressed as requested and figures have been added to the manuscript. The English will need to be improved in occasional places in the manuscript before further publication.
Comments on the Quality of English LanguageEnglish remains to be corrected in occasional places but this review is overall easy to read.